# Incidence of cervical, breast and colorectal cancers between 2010 and 2015 in people living with HIV in France

Teresa Rojas Rojas[1☯], Isabelle Poizot-Martin[1,2☯], David Rey[3], Claudine Duvivier[4,5,6], Firouzé Bani-Sadr[7], André Cabie[8], Pierre Delobel[9], Christine Jacomet[10], Clotilde Allavena[11], Tristan Ferry[12], Pascal Pugliese[13], Marc-Antoine Valantin[14,15], Isabelle Lamaury[16], Laurent Hustache-Matthieu[17], Anne Fresard[18], Tamazighth Houyou[2,19], Thomas Huleux[20], Antoine Cheret[21,22], Alain Makinson[23], Véronique Obry-Roguet[1], Caroline Lions[1], Maria Patrizia Carrieri[2,19]*, Camelia Protopopescu[2,19], the Dat'AIDS Study Group[¶]

1 Aix-Marseille Univ, APHM Sainte-Marguerite, Clinical Immuno-Hematological Unit Marseille, Marseille, France, 2 Aix-Marseille Univ, INSERM, IRD, SESSTIM, Sciences Économiques & Sociales de la Santé & Traitement de l'Information Médicale, ISSPAM, Marseille, France, 3 Le Trait d'Union, HIV-Infection Care Center, Hôpitaux Universitaires de Strasbourg, Strasbourg, France, 4 APHP-Hôpital Necker-Enfants Malades, Service de Maladies Infectieuses et Tropicales, Centre d'Infectiologie Necker-Pasteur, IHU Imagine, Paris, France, 5 Institut Cochin—CNRS 8104—INSERM U1016—RIL Team: Retrovirus, Infection and Latency, Université de Paris, Paris, France, 6 Centre Médical de l'Institut Pasteur, Institut Pasteur, Paris, France, 7 Department of Internal Medicine, Clinical Immunology and Infectious Diseases, Robert Debré Hospital, University Hospital, Reims, France, 8 Université des Antilles, CHU de Martinique, Fort-de-France, Martinique, France, 9 CHU de Toulouse, Service des Maladies Infectieuses et Tropicales-INSERM, UMR 1043- Université Toulouse III Paul Sabatier, Toulouse, France, 10 Clermont-Ferrand University Hospital Infectious and Tropical disease Department, Clermont Ferrand, France, 11 Infectious Diseases Department, CHU Hôtel-Dieu, INSERM UIC 1413, CHU Nantes, Nantes, France, 12 Service de Maladies Infectieuses, Hospices Civils de Lyon, Université Claude Bernard Lyon 1, Villeurbanne, France, 13 Université Côte d'Azur, CHU de Nice, Nice, France, 14 GHPS Pitié Salpêtrière APHP, Infectious Diseases, Paris, France, 15 Sorbonne Universités UPMC Université Paris 6-INSERM-IPLESP, Paris, France, 16 Department of Infectious and Tropical Diseases, University Hospital of Pointe-à-Pitre, Pointe-à-Pitre, France, 17 Department of Infectious and Tropical Diseases, Jean Minjoz University Hospital, Besançon, France, 18 Department of Infectious and Tropical Diseases, University Hospital of Saint-Etienne, Saint-Etienne, France, 19 ORS PACA, Observatoire Régional De La Santé Provence-Alpes-Côte d'Azur, Marseille, France, 20 Service Universitaire des Maladies Infectieuses et du Voyageur—Centre Hospitalier G. DRON Tourcoing, Tourcoing, France, 21 Université Paris Descartes, Sorbonne Paris Cité, Paris, France, 22 Department of Internal Medicine, Bicêtre Hospital, AP-HP, Le Kremlin-Bicêtre, France, 23 Department of Infectious Diseases, Montpellier University Hospital, INSERM U1175/IRD UMI 233, Montpellier, France

☯ These authors contributed equally to this work.
¶ A list of other contributing authors is given in the Acknowledgments.
* pmcarrieri@aol.com

**Data Availability Statement:** All relevant data are within the paper and its Supporting Information files.

## Abstract

### Background

We aimed to evaluate the incidence rates between 2010 and 2015 for invasive cervical cancer (ICC), breast cancer (BC), and colorectal cancer (CRC) in people living with HIV (PLWH) in France, and to compare them with those in the French general population. These cancers are targeted by the national cancer-screening program.

### Setting

This is a retrospective study based on the longitudinal data of the French Dat'AIDS cohort.

**Funding:** The author(s) received no specific funding for this work.

**Competing interests:** The authors have declared that no competing interests exist.

## Methods

Standardized incidence ratios (SIR) for ICC and BC, and incidence rates for all three cancers were calculated overall and for specific sub-populations according to nadir CD4 cell count, HIV transmission category, HIV diagnosis period, and HCV coinfection.

## Results

The 2010–2015 CRC incidence rate was 25.0 [95% confidence interval (CI): 18.6–33.4] per 100,000 person-years, in 44,642 PLWH (both men and women). Compared with the general population, the ICC incidence rate was significantly higher in HIV-infected women both overall (SIR = 1.93, 95% CI: 1.18–3.14) and in the following sub-populations: nadir CD4 $\leq$ 200 cells/mm$^3$ (SIR = 2.62, 95% CI: 1.45–4.74), HIV transmission through intravenous drug use (SIR = 5.14, 95% CI: 1.93–13.70), HCV coinfection (SIR = 3.52, 95% CI: 1.47–8.47) and HIV diagnosis before 2000 (SIR = 2.06, 95% CI: 1.07–3.97). Conversely, the BC incidence rate was significantly lower in the study sample than in the general population (SIR = 0.56, 95% CI: 0.42–0.73).

## Conclusion

The present study showed no significant linear trend between 2010 and 2015 in the incidence rates of the three cancers explored in the PLWH study sample. Specific recommendations for ICC screening are still required for HIV-infected women and should focus on sub-populations at greatest risk.

## Introduction

Cancer is on track to soon become the leading cause of death in PLWH in France [1]. While expanded access to antiretroviral treatment (ART) has resulted in an impressive decline in AIDS-defining cancers (ADC) [2–4] in people living with HIV (PLWH) thanks to greater immune recovery and reduced HIV viral replication [5], the risk of non-AIDS-defining cancers (NADC) is still higher in PLWH—including people with long-term viral suppression [3, 6–8]—than in the general population.

Increased PLWH life expectancy—thanks to the widespread use of ART [9, 10]—together with exposure to other oncogenic viruses (e.g., human papilloma virus (HPV), hepatitis C virus (HCV), hepatitis B virus (HBV) and the Epstein-Barr virus [11]), may partly explain the increased incidence in NADC [3, 12] and their predominance in terms of cancer morbidity and mortality in PLWH [13–16]. Using longitudinal data from the French Dat'AIDS cohort, we recently reported that NADC incidence between 2010 and 2015 was higher than for ADC, with the former representing 76.6% of incident cancer cases [17].

National screening for identification of pre-cancerous lesions is one key to reducing cancer incidence and mortality worldwide [18]. In France, invasive cervical cancer (ICC), breast cancer (BC) and colorectal cancer (CRC) are the three cancers targeted by the national cancer-screening program. However, in the latest expert report on the medical management of PLWH in France [19], specific screening guidelines are only provided for ICC. These recommend annual screening for women living with HIV (WLWH). This differs from the three-yearly screening recommendation in France for women in the general population.

ICC, which is associated with HPV infection, is the fourth most frequently diagnosed cancer and the fourth leading cause of cancer mortality in women worldwide [20]. For both sexes combined, BC was the second most common cancer and the fifth cause of death from cancer worldwide in the general population in 2018 [20]. For women, it is the leading cause of cancer mortality and the cancer most commonly diagnosed [20]. CRC has the third highest incidence of any cancer worldwide for both sexes combined, and is second only to lung cancer in terms of mortality [20].

This study aimed to estimate the incidence rates of these three cancers between 2010 and 2015 using data from the large longitudinal French PLWH cohort Dat'AIDS, and to compare them with the related incidence rates for the French general population.

## Materials and methods

### Data source

A retrospective analysis was performed using longitudinal data from the French multicenter Dat'AIDS cohort (NCT 02898987 ClinicalTrials.gov), which was created in 2010 as a collaboration between 17 major French HIV clinical centers that were already using a common electronic medical record (NADIS®). The aim of the cohort was to constitute a compilation of databases for the follow-up of individuals infected with HIV, HBV and/or HCV [21]. Data collection was approved by the French National Commission on Informatics and Liberty (CNIL 2001/762876). All patients provided signed informed consent before inclusion. Clinical and patient-related data were recorded during medical encounters in a structured database, with data quality ensured by automated checks during data capture, regular controls, and *ad hoc* processes before any scientific analysis was performed.

### Study population and study period

The present study population included 44,642 HIV-infected patients aged 15 years and over, with or without a history of cancer, and having at least one follow-up visit in the Dat'AIDS cohort between 01 January 2010 and 31 December 2015. The study period began on 01 January 2010 for patients included in Dat'AIDS before this date, and on the database inclusion date for those included after 01 January 2010. For each of the three study analyses (one for each cancer type), the end of the study period was set at the first diagnosis of the specific cancer type for patients with at least one diagnosis of that cancer between 01 January 2010 and 31 December 2015, and at the last database update for all other patients, with censoring at 31 December 2015. The ICC and BC analyses were restricted to women only.

### Definition of cancer cases

The three cancer types were defined according to the International Classification of Diseases, Tenth Revision (ICD-10) [22]. Specifically, the following ICD-10 codes were used: C18-C20 for CRC, C50 for BC and C53 for ICC. For each cancer type, only the first diagnosis during the study period was retained in the analysis. Prevalent cancer cases—defined here as diagnosis during the first 30 days after enrolment in Dat'AIDS—were excluded. Recurring and metastatic cancers were also excluded.

### Descriptive variables

The following variables were used to describe the study population, both overall and according to the three cancer types: sex, age, HIV transmission mode (heterosexual, intravenous drug use (IVDU), men who have sex with men (MSM), and other), time since HIV diagnosis, HIV

diagnosis period (before or after 2000), CDC HIV stage (A/B or C), HCV coinfection (defined by HCV-positive serology, nadir CD4 cell count, current CD4 cell count and HIV plasma viral load (HIV-pVL)), length of time with undetectable viral load (HIV-pVL $\leq$ 50 copies/mL), being ART naive, ART regimen (nucleotide reverse transcriptase inhibitor (NRTI), nonnucleoside reverse transcriptase inhibitor (NNRTI), protease inhibitor (PI) or integrase strand transfer inhibitor (INSTI)), first-line ART, length of time on ART since initiation, length of time on most recent line of ART, and history of cancer. All these variables were measured at the first diagnosis of each cancer type, or at the last available visit for patients with no cancer diagnosed during the study period.

## Statistical analysis

Incidence rates per 100,000 person-years for each cancer type were calculated globally for the entire study population, and separately for men and women for CRC. Specifically, incidence rates were calculated as the number of new cases occurring during the study period divided by the total follow-up time for people at risk, i.e. the number of person-years.

We compared the incidence rates with those for the French general population. For the latter, we used the 2012 values in the Francim network report [23], assuming that rates were constant between 2010 and 2015 at the national level. Specifically, we calculated the standardized incidence ratios (SIR) (see below) for ICC and BC, using indirect standardization [24], and their 95% confidence intervals based on a Poisson distribution. The SIR for CRC could not be calculated, because incidence rate data for the French general population were lacking (they were available only for colon, rectum, and anal cancers grouped together, but not for CRC without anal cancer). Data were aggregated by sex and age strata, using the following 5-year age intervals: [15–19], [20–24], [25–29], [30–34], [35–39], [40–44], [45–49], [50–54], [55–59], [60–64], [65–69], [70–74], [75–79], [80–84], [85–89], [90–94] and [$\geq$ 95]. Incidence rates were computed for each sex-age stratum and compared with the related observed incidence rates in the general population.

The SIR represented the ratio between the incidence rate in our cohort and the incidence rate in the general population, after adjusting for sex and age. If the SIR was >1, then an excess risk of cancer existed in our cohort compared to the general population. More specifically, the incidence rate in the cohort was (SIR-1)*100 percent higher than in the general population. Conversely, if SIR was <1, then the incidence rate in the cohort was (1-SIR)*100 percent lower than in the general population.

The incidence rates and SIR were computed globally and according to nadir CD4 cell count, HIV CDC stage, HIV transmission mode, HIV diagnosis period, and HCV coinfection. We also computed the incidence rates per calendar year, and performed a $\chi^2$ test for linear trend to determine whether a significant trend occurred in new cancer cases over the study period. A p-value <0.05 was considered significant.

All analyses were performed with Stata 14.2 software for Windows.

## Results

### Characteristics of the study population

Of the 44,642 PLWH followed up in the Dat'AIDS cohort during the study period, 13,543 were women (30.3%) representing 180,216.4 person-years. The characteristics of the study population are provided in Table 1. Median time since HIV diagnosis was 13 years [interquartile range (IQR): 6–21]. For each of the three cancer types studied, median time since HIV diagnosis was higher in patients with that cancer than in the whole study population. Thirty-eight percent of the study population were MSM, 8.2% were HIV infected through IVDU and 15%

**Table 1. Main characteristics of the study population (HIV-infected patients followed up in the Dat'AIDS cohort between January 1, 2010 and December 31, 2015) (N = 44,642).**

| | Study population | Women with ICC | Women with BC | Patients with CRC |
|---|---|---|---|---|
| | N = 44,642 | N = 16 | N = 51 | N = 45 |
| **Characteristics of patients*** | No. of patients (%) or median [IQR] | No. of patients (%) or median [IQR] | No. of patients (%) or median [IQR] | No. of patients (%) or median [IQR] |
| **Sex** | | | | |
| Female | 13,543 (30.3) | 16 (100.0) | 51 (100.0) | 10 (22.2) |
| Male | 31,099 (69.6) | – | – | 35 (77.7) |
| **Age (years)** | 48 [40–55] | 49 [45–55] | 47 [42–57] | 57 [50–63] |
| **Time since HIV diagnosis (years)** | 13 [6–21] | 15 [8–22] | 16 [8–22] | 20 [13–23] |
| **HIV diagnosis period** | | | | |
| After 2000 | 24,740 (55.4) | 7 (43.8) | 22 (43.1) | 11 (24.4) |
| Before 2000 | 19,902 (44.6) | 9 (56.2) | 29 (56.9) | 34 (75.6) |
| **HIV transmission mode** | | | | |
| Heterosexual | 18,979 (42.9) | 10 (62.5) | 39 (76.4) | 15 (33.3) |
| IVDU | 3,631 (8.2) | 4 (25.0) | 4 (7.8) | 6 (13.3) |
| MSM | 16,986 (38.4) | – | – | 18 (40.0) |
| Other | 4,623 (10.4) | 2 (12.5) | 8 (15.6) | 6 (13.3) |
| **HIV CDC stage** | | | | |
| A | 26,879 (61.0) | 0 (0.0) | 28 (54.9) | 13 (28.8) |
| B | 6,801 (15.4) | 0 (0.0) | 13 (25.4) | 10 (22.2) |
| C | 10,355 (23.5) | 16 (100.0) | 10 (19.6) | 22 (48.8) |
| **HCV coinfection** | | | | |
| No | 37,919 (84.9) | 11 (68.7) | 44 (86.2) | 35 (77.7) |
| Yes | 6,723 (15.0) | 5 (31.2) | 7 (13.7) | 10 (22.2) |
| **Nadir CD4 cell count/mm³** | 223 [97–350] | 122 [42–245] | 231 [82–326] | 82 [17–190] |
| >200 | 23,984 (54.9) | 4 (26.7) | 31 (60.7) | 11 (24.4) |
| ≤200 | 19,690 (45.1) | 11 (73.3) | 20 (39.2) | 34 (75.6) |
| **CD4 cell count/mm³** | 601 [413–810] | 354 [264–627] | 718 [408–943] | 402 [271–630] |
| ≤200 | 3,168 (7.2) | 3 (21.4) | 0 (0.0) | 5 (11.1) |
| 201–500 | 12,587 (28.9) | 7 (50.0) | 17 (33.3) | 22 (48.8) |
| >500 | 27,778 (63.8) | 4 (28.5) | 34 (66.6) | 18 (40.0) |
| **HIV-pVL (copies/mL)** | 20 [20–40] | 30 [20–40] | 20 [20–40] | 20 [20–40] |
| **HIV-pVL (copies/mL)** | | | | |
| ≤50 | 36,181 (83.1) | 12 (85.7) | 45 (88.2) | 39 (86.6) |
| 51–1000 | 3,492 (8.0) | 1 (7.1) | 3 (5.8) | 5 (11.1) |
| 1001–10000 | 1,358 (3.1) | 0 (0.0) | 1 (1.9) | 1 (2.2) |
| 10001–100000 | 1,665 (3.8) | 1 (7.1) | 2 (3.9) | 0 (0.0) |
| >100000 | 830 (1.9) | 0 (0.0) | 0 (0.0) | 0 (0.0) |
| **Time with undetectable viral load (HIV-pVL <50 copies/mL) (years)** | 2.5 [0.4–6.0] | 1.4 [0.3–3.7] | 2.8 [0.5–6.2] | 2.5 [0.5–6.2] |
| **ART naive** | | | | |
| No | 40,670 (91.1) | 16 (100.0) | 49 (96.0) | 45 (100.0) |
| Yes | 3,972 (8.9) | 0 (0.0) | 2 (3.9) | 0 (0.0) |
| **Antiretroviral regimen** | | | | |
| 2 NRTI+1 boosted-PI | 11,644 (28.6) | 7 (50.0) | 16 (34.7) | 15 (33.3) |
| 2 NRTI+1 NNRTI | 13,681 (33.6) | 1 (7.1) | 15 (32.6) | 13 (28.8) |
| 2 NRTI+1 INSTI | 7,445 (18.3) | 2 (14.2) | 6 (13.0) | 5 (11.1) |
| Other | 7,893 (19.4) | 4 (28.5) | 9 (19.5) | 12 (26.6) |
| **On first-line ART** | | | | |

(Continued)

**Table 1.** (Continued)

| | Study population | Women with ICC | Women with BC | Patients with CRC |
|---|---|---|---|---|
| | N = 44,642 | N = 16 | N = 51 | N = 45 |
| No | 34,101 (83.8) | 13 (92.8) | 39(84.7) | 40 (88.8) |
| Yes | 6,569 (16.1) | 1 (7.1) | 7 (15.2) | 5 (11.1) |
| **Length of time on ART since initiation** (*years*) | 9 [3–17] | 10 [5–15] | 12 [6–17] | 14 [11–18] |
| **Length of time on most recent line of ART** (*months*) | 21 [6–52] | 16 [8–23] | 23 [9–54] | 25 [9–49] |
| **History of cancer** | | | | |
| No | 41,897 (93.8) | 15 (93.7) | 44 (86.2) | 31 (68.8) |
| Yes | 2,745 (6.1) | 1 (6.2) | 7 (13.7) | 14 (31.1) |

ART: antiretroviral treatment, BC: breast cancer; ICC: invasive cervical cancer; IQR: interquartile range; MSM: men who have sex with men; HIV-pVL: HIV plasma viral load; HCV: hepatitis C virus; IVDU: intravenous drug use; NRTI: nucleotide reverse transcriptase inhibitor; PI: protease inhibitor; NNRTI: nonnucleoside reverse transcriptase inhibitor; INSTI: integrase strand transfer inhibitor.

* These variables were measured at the first diagnosis of each cancer type, or at the last available visit for patients with no cancer diagnosed during the study period.

were HCV coinfected. HIV transmission through IVDU was over-represented in women with ICC, while heterosexual HIV transmission mode was under-represented in men and women with CRC.

Median CD4 cell count and nadir CD4 cell count at the first diagnosis of each cancer type, or at the last available visit for patients with no cancer diagnosed during the study period, were 601 [IQR: 413–810] and 223 [IQR: 97–350] cells/mm$^3$, respectively. Median CD4 cell count at the first diagnosis of each cancer type was 354 [IQR: 264–627] for patients with ICC, 718 [IQR: 408–943] for patients with BC and 402 [IQR: 271–630] for patients with CRC (Table 1). Among the study population, median length of time on ART was 9 years [IQR: 3–17]. Finally, 83.1% of patients had an HIV-pVL < 50 copies/mL and 8.9% were ART naive.

## Cancer cases

A total of 1,454 patients were diagnosed with cancer during the follow-up period. Of these, 140 had a prevalent cancer. Among the remaining 1,314 incident cases, the following were retained for the analyses: 16 ICC, 45 CRC and 55 BC. Four of the latter were in men; these were excluded from the analysis, as only women are targeted by the French national cancer-screening program, and data concerning men were not available for the general population.

## ICC, BC and CRC incidence rates, both overall and according to patient profile

Among the 13,543 women included, accounting for 56,150.6 person-years, the overall incidence rates between 2010–2015 of BC and ICC were 91.0 [95% CI: 69.1–119.7] and 28.5 [95% CI: 17.5–46.5] per 100,000 person-years, respectively (Table 2). The CRC incidence rate in the whole study population was 25.0 [95% CI: 18.7–33.5] per 100,000 person-years overall, specifically, 28.3 [95% CI: 20.3–39.4] and 17.8 [95% CI: 9.6–33.2] per 100,000 person-years in men and women, respectively. The incidence rates per calendar year showed a non-significant linear trend over the study period for the three different cancers (p-value = 0.071, 0.237, and 0.672, for BC, ICC, and CRC, respectively, Fig 1A–1C).

In terms of patient profile, incidence rates of all three cancers were higher in patients diagnosed with HIV before 2000. The ICC and CRC incidence rates were higher in those with nadir CD4 ≤ 200 cells/mm$^3$, and in women with HCV coinfection. Conversely, the BC

**Table 2. Incidence rates of cervical, breast and colorectal cancers—overall and according to patient profile—in the Dat'AIDS cohort between 01 January 2010 and 31 December 2015 (N = 44,642).**

| | BC (Women) | | ICC (Women) | | CRC (All) | | CRC (Men) | | CRC (Women) | |
|---|---|---|---|---|---|---|---|---|---|---|
| | N | IR [95% CI] | N | IR [95% CI] | N | IR [95% CI] | N | IR [95% CI] | N | IR [95% CI] |
| **Total** | 51 | 91.0 [69.1–119.7] | 16 | 28.5 [17.5–46.5] | 45 | 25.0 [18.7–33.5] | 35 | 28.3 [20.3–39.4] | 10 | 17.8 [9.6–33.2] |
| **HIV diagnosis period** | | | | | | | | | | |
| Before 2000 | 29 | 108.9 [75.7–156.7] | 9 | 33.8 [17.6–65.0] | 34 | 37.4 [26.8–52.4] | 24 | 37.4 [25.1–55.8] | 10 | 37.6 [20.2–69.8] |
| After 2000 | 22 | 74.5 [49.1–113.2] | 7 | 23.7 [11.3–49.7] | 11 | 12.3 [6.8–22.2] | 11 | 18.4 [10.2–33.2] | 0 | 0 |
| **HIV CDC stage** | | | | | | | | | | |
| A/B | 41 | 94.2 [69.4–128.0] | | _ | 23 | 17.1 [11.4–25.8] | 19 | 20.9 [13.3–32.8] | 4 | 9.2 [3.4–24.5] |
| C | 10 | 79.7 [42.9–148.2] | | _ | 22 | 48.4 [31.8–73.4] | 16 | 48.6 [29.7–79.3] | 6 | 47.8 [21.5–106.5] |
| **Nadir CD4 cell count/mm³** | | | | | | | | | | |
| ≤200 | 20 | 73.0 [47.1–113.1] | 11 | 40.1 [22.2–72.5] | 34 | 38.6 [27.6–54.1] | 26 | 42.9 [29.2–63.0] | 8 | 29.2 [14.6–58.4] |
| >200 | 31 | 108.7 [76.4–154.5] | 4 | 14.0 [5.3–37.4] | 11 | 12.0 [6.7–21.7] | 9 | 14.3 [7.4–27.5] | 2 | 7.0 [1.8–28.0] |
| **HIV transmission mode** | | | | | | | | | | |
| Heterosexual | 39 | 86.5 [63.2–118.4] | 10 | 22.2 [11.9–41.2] | 15 | 19.0 [11.4–31.5] | 9 | 26.5 [13.8–51.0] | 6 | 13.3 [6.0–29.6] |
| IVDU | 4 | 90.3 [33.9–240.7] | 4 | 90.3 [33.9–240.7] | 6 | 37.5 [16.8–83.5] | 4 | 34.6 [13.0–92.1] | 2 | 45.2 [11.3–180.6] |
| MSM | _ | _ | _ | _ | 18 | 26.5 [16.7–42.0] | 18 | 26.6 [16.8–42.2] | _ | _ |
| Other | 8 | 127.8 [63.9–255.5] | 2 | 31.9 [8.0–127.7] | 6 | 35.5 [15.9–79.0] | 4 | 37.6 [14.1–100.1] | 2 | 31.9 [8.0–127.7] |
| **HCV coinfection** | | | | | | | | | | |
| No | 44 | 92.3 [68.7–124.0] | 11 | 23.1 [12.8–41.7] | 35 | 23.2 [16.7–32.3] | 30 | 29.1 [20.4–41.7] | 5 | 10.5 [4.4–25.2] |
| Yes | 7 | 82.7 [39.4–173.5] | 5 | 59.1 [24.6–142.0] | 10 | 33.9 [18.2–62.9] | 5 | 23.7 [9.9–57.0] | 5 | 59.1 [24.6–142.0] |

BC: breast cancer; CI: confidence interval; CRC: colorectal cancer; HCV: hepatitis C virus; ICC: Invasive cervical cancer; IR: incidence rate per 100,000 person-years; IVDU: intravenous drug use; MSM: men who have sex with men.

incidence rates were lower in these sub-populations (Table 2). However, we could not conclude about statistical significance of the differences in incidence rates between any sub-populations, as the confidence intervals overlapped in most cases.

## SIR for ICC and BC, overall and according to patient profile

The ICC incidence rate in the women in our study population was 93% higher (SIR = 1.93; 95% CI, 1.18–3.14) than that in women in the general population (Table 3). In terms of patient profile, ICC incidence was significantly higher in the following sub-populations of WLWH than in the general population: nadir CD4 ≤ 200 cells/mm³ (SIR = 2.62, 95% CI: 1.45–4.74), HIV-infected through IVDU (SIR = 5.14, 95% CI: 1.93–13.70), HIV diagnosis before 2000 (SIR = 2.06, 95% CI: 1.07–3.97), and HCV coinfected (SIR = 3.52, 95% CI: 1.47–8.47).

The incidence rate observed for BC was 44% lower (SIR = 0.56, 95% CI: 0.42–0.73) overall in the study population than in the general population, and in all women sub-populations (Table 3).

## Discussion

This study evaluated the incidence rates for the period 2010–2015 for ICC, BC and CRC, which are the three cancers included in the French national screening program for cancer prevention. It highlighted that: (i) the ICC incidence rate was 93% higher in WLWH than in women in the general population, with significant differences according to HIV-related characteristics; (ii) the BC incidence rate was 44% lower in WLWH than in the general population;

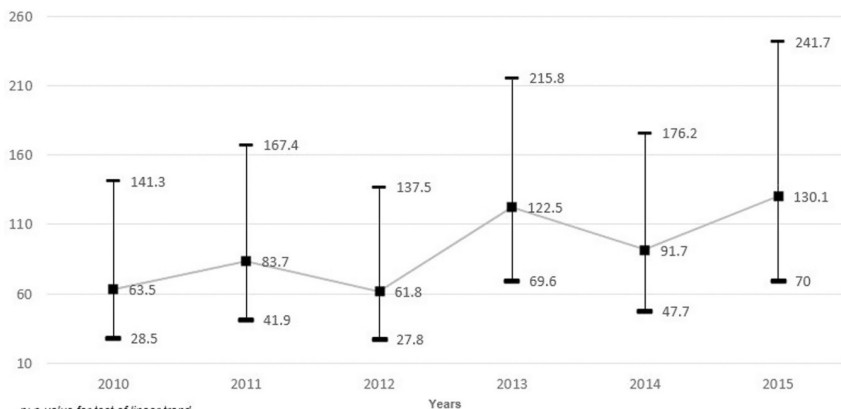

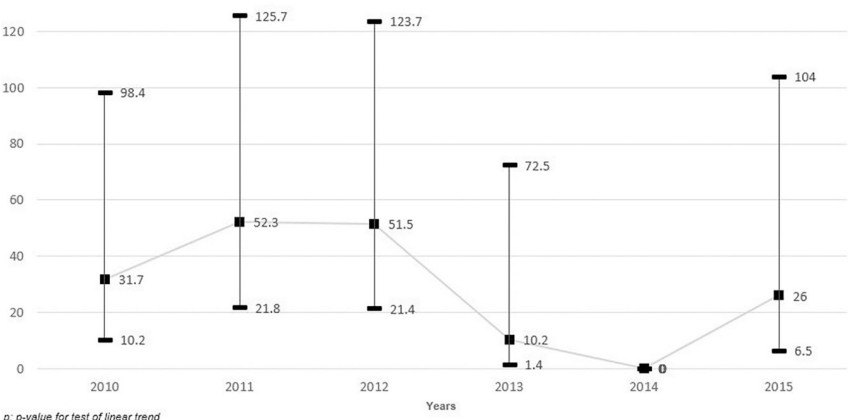

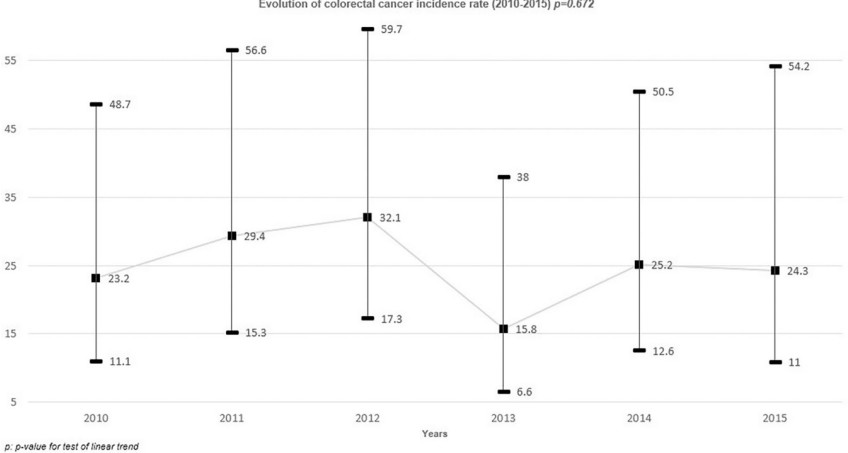

**Fig 1. Evolution of cancer incidence rates by calendar year in the Dat'AIDS cohort between 01 January 2010 and 31 December 2015.** A. Breast cancer. B. Cervical cancer. C. Colorectal cancer.

**Table 3. Standardized incidence ratios of breast cancer and invasive cervical cancer in women in the Dat'AIDS cohort between January 1, 2010 and December 31, 2015 (N = 13,543).**

| | BC | | ICC | |
| | N = 51 | | N = 16 | |
| | No. of cancer cases | SIR [95% CI] | No. of cancer cases | SIR [95% CI] |
|---|---|---|---|---|
| **Total** | 51 | 0.56 [0.42–0.73] | 16 | 1.93 [1.18–3.14] |
| **HIV diagnosis period** | | | | |
| Before 2000 | 29 | 0.55 [0.38–0.78] | 9 | 2.06 [1.07–3.97] |
| After 2000 | 22 | 0.57 [0.38–0.87] | 7 | 1.77 [0.84–3.72] |
| **Nadir CD4 cell count/mm$^3$** | | | | |
| ≤200 | 20 | 0.42 [0.27–0.65] | 11 | 2.62 [1.45–4.74] |
| >200 | 31 | 0.71 [0.50–1.01] | 4 | 0.98 [0.37–2.61] |
| **HIV transmission mode** | | | | |
| Heterosexual | 39 | 0.54 [0.39–0.74] | 10 | 1.50 [0.81–2.79] |
| IVDU | 4 | 0.44 [0.16–1.17] | 4 | 5.14 [1.93–13.70] |
| Other | 8 | 0.83 [0.42–1.66] | 2 | 2.48 [0.62–9.93] |
| **HCV coinfection** | | | | |
| No | 44 | 0.59 [0.44–0.79] | 11 | 1.60 [0.88–2.88] |
| Yes | 7 | 0.42 [0.20–0.87] | 5 | 3.52 [1.47–8.47] |

BC: breast cancer; ICC: invasive cervical cancer; CI: confidence interval; HCV: hepatitis C virus; IVDU: intravenous drug use; SIR: standardized incidence ratio.

(iii) the incidence rates of the three studied cancers did not show a significant linear trend between 2010 and 2015.

The higher ICC incidence in PLWH than in the general population echoes findings from previous studies [25, 26] and is in line with data from the French Hospital Database on HIV (FHDH) until 2009, reported by Grabar et al. [27]. ICC incidence in the latter study was more than three times higher in PLWH than in the general population [27] and the SIR of ICC was consistently >1 in all the periods considered: pre ART (1992–1996), early ART (1997–2000), intermediate ART (2001–2004) and late ART (2005–2009) [27]. Our study showed that while most PLWH in France between 2010 and 2015 were on ART with controlled HIV viremia, the ICC incidence rate was still significantly higher in WLWH than in the general population. However, the rate we observed was lower than that reported by Grabar et al. [27]. Our study design did not allow us to determine whether this difference was related to a change in health-care access or in ICC screening uptake. Moreover, the incidence rates and SIR of ICC were both higher in the following sub-populations of women (although we could not state on the statistical significance of these differences): HIV transmission through IVDU, HCV coinfected, nadir CD4 < 200 cells/mm$^3$, and diagnosed with HIV before 2000. Accordingly, in the recent ART era, the most vulnerable WLWH had the higher risk of ICC, stressing the need to further evaluate the efficacy of HPV vaccines in reducing ICC incidence [28]. The lack of data on history of HPV infection, whose causal role in the ICC has been proven [29], prevented us from including this information in our study.

In the French general population, the incidence rate for CRC together with anal cancer in 2012 was estimated at 38.4 per 100,000 person-years in men and 23.7 per 100,000 person-years in women, with a decrease of 0.3% per year between 2005–2012 in both sexes [30]. In our cohort, this decrease was not observed for CRC between 2010–2015. Unfortunately, due to the lack of information on CRC only (i.e., without anal cancer) in the general population, we could not compare our data with the general French population. However, a retrospective matched cohort study performed between 1999 and 2007 found that the rate of CRC in PLWH

was similar to that in persons not infected with HIV after adjusting for comorbidity variables [31].

With regard to BC, the women in our study (both overall and in terms of each specific sub-population) had a lower incidence rate than women in the general population. The lower incidence rate in WLWH in our study is consistent with previous studies [32]. One hypothesis for this is that ART targeting both viral and tumorigenic proteins may interfere with oncogenic pathways, reducing tumour progression [33]. Another hypothesis is that cellular apoptosis is produced due to HIV binding to the receptor CXCR4 on breast epithelium [34]. Among a total of 31,099 men in our study, only four BC cases were observed (these cases were not included in the incidence analyses). Due to very low incidence, BC screening awareness in men remains poor [35], which might lead to late diagnosis [36, 37].

The higher incidence of BC observed in some sub-populations of WLWH may be explained, at least partially, by the HIV-related characteristics (nadir CD4 > 200 cells/mm$^3$, HIV transmission through IVDU, HIV diagnosis before 2000) and absence of HCV coinfection.

In the general population, chronic HCV infection has not only been other associated with an increased risk of hepatocellular carcinoma, but also of other cancers (e.g., oesophagus, pancreas, prostate, thyroid, breast and oral cavity) [38, 39]. In our cohort, the ICC incidence rate and SIR, as well as the incidence rate of CRC were higher in HCV coinfected women than in women with no HCV coinfection. This was not observed for BC. This result could not be explored in detail due to the lack of HCV RNA data in our study. However, it may be associated with the fact that specific sub-populations, for example drug users, have vulnerabilities associated with an increased risk of HCV infection.

The relationship we found between lower nadir CD4 cell count and higher ICC incidence in our study population echoes the findings in previous studies [27, 40]. Furthermore, our results are in line with other studies which showed a relationship between low CD4 cell count and higher ICC incidence [41–43]. Access to national cancer screening programs may explain these two results. More specifically, with regard to screening access in France, the VESPA study highlighted that a low educational level, irregular gynecological follow-up, and a low CD4 nadir were all barriers to BC and ICC screening access [44]. Furthermore, the rate of BC screening in the previous two years in the VESPA study was lower in WLWH than in the general population (82.2% versus 88%, respectively), and only 39.4% of PLWH (both sexes combined) declared having a fecal occult blood test in the previous two years. However in the same study, the rate of cervical cancer screening was higher in HIV-infected women (88.1%) than in women in the general population (83.1%) [44], which suggests that specific recommendations for PLWH may improve screening uptake and reduce disparities in access to screening [44]. Nevertheless, this hypothesis should be considered with caution, as only 76% of WLWH reported having a Pap smear for cervical cancer screening in the year before the survey, whereas French guidelines at the time of the VESPA study recommended cervical cancer screening each year for HIV-positive women.

Currently, the screening methods for cervical cancer in France differ according to the age of the women. The HPV test, which looks for cervical infection by high-risk types of HPV, is now recommended instead of the Pap test every five years after a first negative result in women aged 30 to 65 years old [45]. Global elimination of cervical cancer is a World Health Organization (WHO) priority, and screening for ICC must be part of a comprehensive preventive approach that includes HPV vaccination, the early treatment of pre-cancerous lesions, and the appropriate and rapid management of women diagnosed with ICC [46].

The main strength of our study was its large sample size, which provided strong statistical power. Moreover, patients were actively enrolled by medical doctors and research technicians. These medical staff were also responsible for data entry on the electronic medical record NADIS®, thereby ensuring data quality.

The study also has several limitations. First, we could not compute SIR for CRC due to the lack of data for the French general population at the time of the analysis. Second, we had no information about the number of patients who had been screened for BC, ICC and CRC in our cohort. Third, data on sociodemographic characteristics (e.g., education level) and behaviors (e.g., tobacco, alcohol and drug consumption) were lacking, which prevented us from being able to evaluate the association between these factors and cancer incidence.

In conclusion, the present study did not show any significant linear trend between 2010 and 2015 in the incidence rates of the three cancer types explored in PLWH. Furthermore, it confirmed previous results that WLWH do not require specific recommendations for BC screening, and highlighted that WLWH had higher ICC incidence rates in specific sub-populations.

Our findings underline the need to maintain specific recommendations for ICC screening in HIV-infected women. National recommendations need to target specific sub-populations of women at risk of ICC, and clinicians in HIV clinical centers should facilitate regular gynecological follow-up for them.

## Supporting information

**S1 Data.**
(XLS)

## Acknowledgments

We thank all the members of the Dat'AIDS Study Group. We especially thank all the physicians and nurses involved in the follow-up of the cohort and all the patients who took part in this study. Finally, we thank Jude Sweeney (Milan, Italy) for the English revision and copyediting of our manuscript.

**Dat'AIDS study group:** C. Drobacheff-Thiébaut, A. Foltzer, K. Bouiller, L. Hustache-Mathieu, C. Chirouze, Q. Lepiller, F. Bozon, O Babre, A.S. Brunel, P. Muret (*Besançon*); H. Laurichesse, O. Lesens, M. Vidal, N. Mrozek, C. Aumeran, O. Baud, V. Corbin, P. Letertre-Gibert, S. Casanova, J. Prouteau, C. Jacomet (*Clermont Ferrand*); I. Lamaury, I. Fabre, E. Curlier, R. Ouissa, C. Herrmann-Storck, B. Tressieres, T. Bonijoly,M.C. Receveur, F. Boulard, C. Daniel, C.Clavel (*Guadeloupe*); D. Merrien, P. Perré, T. Guimard, O. Bollangier, S. Leautez, M. Morrier, L. Laine (*La Roche sur Yon*); F. Ader, A. Becker, F. Biron, A. Boibieux, L. Cotte, T. Ferry, P Miailhes, T. Perpoint, S. Roux, C. Triffault-Fillit, S. Degroodt, C. Brochier, F Valour, C. Chidiac (*Lyon*); A. Ménard, A.Y. Belkhir, P.Colson, C. Dhiver, A.Madrid, M.Martin-Degioanni, L. Meddeb, M. Mokhtari, A.Motte, A.Raoux, I. Ravaux, C.Tamalet, C. Toméi, H. Tissot Dupont (*Marseille IHU Méditerranée*); S. Brégigeon, O. Zaegel-Faucher, V. Obry-Roguet, H Laroche, M. Orticoni, M.J. Soavi, P Geneau de Lamarlière, E Ressiot, M.J. Ducassou, I. Jaquet, S. Galie, A Galinier, P. Martinet, M. Landon, A.S. Ritleng, A. Ivanova, C. Debreux, C. Lions, T. Rojas, I. Poizot-Martin (*Marseille Ste Marguerite*); S. Abel, O. Cabras, L. Cuzin, K. Guitteaud, M. Illiaquer, S. Pierre-François, L. Osei, J. Pasquier, K. Rome, E. Sidani, JM Turmel, C. Varache, A. Cabié (*Martinique*); N. Atoui, M. Bistoquet, E Delaporte, V. Le Moing, A. Makinson, N. Meftah, C. Merle de Boever, B. Montes, A. Montoya Ferrer, E. Tuaillon, J. Reynes (*Montpellier*); M. André, L. Boyer, MP. Bouillon, M. Delestan, C. Rabaud, T. May, B. Hoen (Nancy); C. Allavena, C. Bernaud, E. Billaud, C. Biron, B. Bonnet, S. Bouchez, D. Boutoille, C. Brunet-Cartier, C. Deschanvres, N. Hall, T. Jovelin, P. Morineau, V. Reliquet, S. Sécher, M. Cavellec, A. Soria, E. Paredes, V. Ferré, E. André-Garnier, A. Rodallec, M. Lefebvre, O. Grossi, O. Aubry, F. Raffi (Nantes); P. Pugliese, S. Breaud, C. Ceppi, D. Chirio, E. Cua, P. Dellamonica,

E. Demonchy, A. De Monte, J. Durant, C. Etienne, S. Ferrando, R. Garraffo, C. Michelangeli, V. Mondain, A. Naqvi, N. Oran, I. Perbost, S. Pillet, C. Pradier, B. Prouvost-Keller, K. Risso, V. Rio, PM. Roger, E. Rosenthal, S. Sausse I. Touitou, S. Wehrlen-Pugliese, G. Zouzou (*Nice*); L. Hocqueloux, T. Prazuck, C. Gubavu, A. Sève, A. Maka, C. Boulard, G. Thomas (*Orleans*); A. Cheret, C.Goujard, Y.Quertainmont, E.Teicher, N. Lerolle, O.Deradji, A.Barrail-Tran (*Paris Hop. Bicètre*); R. Landman, V. Joly, J Ghosn, C. Rioux, S. Lariven, A. Gervais, F.X. Lescure, S. Matheron, F. Louni, Z. Julia, C. Mackoumbou-Nkouka, S Le Gac C. Charpentier, D. Descamps, G. Peytavin, Y. Yazdanpanah (*Paris Hop. Bichat*); K. Amazzough, G. Benabdelmoumen, P. Bossi, G. Cessot, C. Charlier, P.H. Consigny, F. Danion, A. Dureault, C. Duvivier, J. Goesch, R. Guery, B. Henry, K. Jidar, F. Lanternier, P. Loubet, O. Lortholary, C. Louisin, J. Lourenco, P. Parize, B. Pilmis, F Touam. (*Paris Hop. Necker Pasteur*); M.A. Valantin, R. Tubiana, R Agher, S.Seang, L.Schneider, R.PaLich, C. Blanc, C. Katlama (*Paris Hop. Pitié Salpétrière*); J.L. Berger, Y. N'Guyen, D. Lambert, I. Kmiec, M. Hentzien, A. Brunet, V. Brodard, F. Bani-Sadr (*Reims*) P. Tattevin, M. Revest, F. Souala, M. Baldeyrou, S. Patrat-Delon, J.M. Chapplain, F. Benezit, M. Dupont, M. Poinot, A. Maillard, C. Pronier, F. Lemaitre, C. Guennoun, M. Poisson-Vanier, T. Jovelin, J.P. Sinteff, C. Arvieux (*Rennes*); E. Botelho-Nevers, A. Gagneux-Brunon, A. Frésard, V. Ronat, F. Lucht (*St Etienne*); P. Fischer, M. Partisani, C Cheneau, M Priester, ML Batard, C Bernard-Henry, E de Mautort, S. Fafi-Kremer, D. Rey (*Strasbourg*); M. Alvarez, N. Biezunski, A. Debard, C. Delpierre, P. Lansalot, L. Lelièvre, G. Martin-Blondel, M. Piffaut, L. Porte, K. Saune, P. Delobel (*Toulouse*); F. Ajana, E. Aïssi, I. Alcaraz, V. Baclet, L. Bocket, A. Boucher, P. Choisy, T. Huleux, B. Lafon-Desmurs, A. Meybeck, M. Pradier, O. Robineau, N. Viget, M. Valette (*Tourcoing*).

## Author Contributions

**Conceptualization:** Teresa Rojas Rojas, Isabelle Poizot-Martin.

**Data curation:** David Rey, Claudine Duvivier, Firouzé Bani-Sadr, André Cabie, Pierre Delobel, Christine Jacomet, Clotilde Allavena, Tristan Ferry, Pascal Pugliese, Marc-Antoine Valantin, Isabelle Lamaury, Laurent Hustache-Matthieu, Anne Fresard, Tamazighth Houyou, Thomas Huleux, Antoine Cheret, Alain Makinson, Véronique Obry-Roguet, Caroline Lions.

**Formal analysis:** Camelia Protopopescu.

**Investigation:** Teresa Rojas Rojas, Isabelle Poizot-Martin.

**Methodology:** Maria Patrizia Carrieri, Camelia Protopopescu.

**Writing – original draft:** Teresa Rojas Rojas, Isabelle Poizot-Martin.

**Writing – review & editing:** Teresa Rojas Rojas, Isabelle Poizot-Martin, David Rey, Claudine Duvivier, Firouzé Bani-Sadr, André Cabie, Pierre Delobel, Christine Jacomet, Clotilde Allavena, Tristan Ferry, Pascal Pugliese, Marc-Antoine Valantin, Isabelle Lamaury, Laurent Hustache-Matthieu, Anne Fresard, Tamazighth Houyou, Thomas Huleux, Antoine Cheret, Alain Makinson, Véronique Obry-Roguet, Caroline Lions, Maria Patrizia Carrieri, Camelia Protopopescu.

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
