## [Decision Letter · Decision Letter 0]

14 Sep 2021

PONE-D-21-14748

Incidence of cervical, breast and colorectal cancer in people living with HIV between 2010 and 2015

PLOS ONE

Dear Dr. Carrieri,

Thank you for submitting your manuscript to PLOS ONE. After careful consideration, we feel that it has merit but does not fully meet PLOS ONE’s publication criteria as it currently stands. Therefore, we invite you to submit a revised version of the manuscript that addresses the points raised during the review process.

Particularly, make sure that you address the comments of the Reviewer 2.

We look forward to receiving your revised manuscript.

Kind regards,

Cristian Apetrei, MD, PhD

Academic Editor

PLOS ONE

Journal Requirements:

Reviewers' comments:

Reviewer's Responses to Questions

**Comments to the Author**

1. Is the manuscript technically sound, and do the data support the conclusions?

Reviewer #1: Yes

Reviewer #2: Partly

2. Has the statistical analysis been performed appropriately and rigorously? 

Reviewer #1: I Don't Know

Reviewer #2: No

3. Have the authors made all data underlying the findings in their manuscript fully available?

Reviewer #1: Yes

Reviewer #2: Yes

4. Is the manuscript presented in an intelligible fashion and written in standard English?

Reviewer #1: Yes

Reviewer #2: No

5. Review Comments to the Author

Reviewer #1: Title: "Incidence of cervical, breast and colorectal cancer in people living with HIV between 2010 and 2015"

The manuscript presents a comprehensive study on a large study sample, regarding incidence trends for the period 2010-2015 for the three cancers included in the French organized screening program for cancer prevention: BC, CRC and ICC.

The study revealed an expected increased of ICC incidence in WLWH than in women in the general population, and also, present an interesting explanation for lower BC incidence rate.

Regarding ICC, it would be worth mentioned that Pap test is not having optimal sensitivity in detecting precancerous lesions of ICC, both in general population and in WLWH.

Organized national CC screening should be implemented worldwide, not only in France, with clinically validated HPV tests, able to detect HR HPV types.

It is known that in WLWH the risk for developing ICC developing is higher, as some studies detected that the risk for evolution to ICC was correlated with the multiple HR HPV types infections and with the number of sexual partners.

Global elimination of cervical cancer is a prioritized aim of WHO, and this could be achieved by two major strategies: stopping the HPV circulation by vaccination and offering HPV screening to those may have been infected before HPV circulation was stopped.

Reviewer #2: Review PONE-D-21-14748

The study has the good intention to provide statistic data for epidemiologic monitoring of cancers among PLWH due to deficient data on cancer incidence in France since 2009. While the study appears to be sound, the language is unclear, making it difficult to follow. The authors should revise the language to improve the flow and readability of the text. Secondly, the study did not clarify how to assign the cancer case that has double or triple cancer diagnosis during analysis. ICC, BC and CRC are the three most common targeted cancers by the nationally organized cancer-screening program in France. Furthermore, BC as the second common cancer in WLWH and the fifth death cause, the study rarely described and discussed BC data. The last but the most important is that the same study using the same data source: the French Dat’AIDS cohort from 2010 to 2015 (Ref 17) have been published in 2020 with much more comprehensive analysis. Altogether, this study will not have a significant contribution to the field.

Minor points to consider in subsequent versions:

1. Significant numbers should be consistent as 0.00 or 0.0 throughout the manuscript.

2. Be consistent with “CD4 cell count” and“HCV coninfection” not “HCV-coinfection” in the text.

3. Some statistic data in text are not match the data listed in Tables. For example, Page 4: Results: The incidence rate of CRC between 2010 and 2015 was 25.0 [95% confidence interval (CI): 18.6-33.4] (in text) vs 25.0 [18.7-33.5] (in Table 2).

4. Tables are better to use the same format. For example, Table 2: 28.5 [17.5-46.5] vs Table 3: 1.93 [1.18 to 3.14].

5. Page 4: Results: Author represented the standardized incidence ratio of ICC and BC, instead of incidence rate in the manuscript.

6. Page6: “the widespread use of ARV[9,10],” should be “the widespread use of ART[9,10],”

7. Page 13: The numbers in the text do not match with the numbers in Table 1: “and nadir CD4 were 601 [IQR: 413-810] and 223 [IQR: 97-350] cells/mm3, respectively. Median CD4 count was 354 [IQR: 264-627] for patients with ICC, 717 [IQR: 408-913] for patients with BC and 446 [IQR: 271-659] for patients with CRC (Table 1).” VS Table 1 numbers: “and nadir CD4 were 223 [97-350] and 601 [413-810]cells/mm3, respectively. Median CD4 count was 354 [264-627] for patients with ICC, 718 [408-943] for patients with BC and 402 [271-630] for patients with CRC. “The incidence rate of CRC in the whole study population was 25.0 [95% CI: 18.6-33.4]” VS Table2: 25.0 [18.7-33.5].

8. Page 14: Table 2: IR for ICC & HCV coinfection is exactly same as IR for CRC (Women) & HCV coinfection: 59.1 [24.6-142.0]. Is it coincident or miss type the data for cancer case?

9. Page 15: Figure 1 has not been explained/presented well in the manuscript. The conclusion “The incidence rates per calendar year remained relatively stable over the study period for the three different cancers (Figs 1A-1C).” from Figure 1 is contradict with Figure 1 presented.

10. Page 15: The section: “SIR for ICC and BC, overall and according to patient profile”: it is supposed to present SIR data from Table 3. However, “incident rate” was in the text instead. Those are two different analysis definition.

11. Page 15: “The ICC incidence rate in the women in our study population was 93% higher (SIR=1.93; 95% CI, 1.18-3.14) than that in women in the general population (Table 3). However, it was 44% lower (SIR, 0.56, 95% CI: 0.42-0.73) for BC.” The calculation/explanation is needed for the results of 93% higher and 44% lower.

12. Page 16: The study claimed that “This is the first study on incidence trends for the period 2010-2015 for the three cancers included in the French organized screening program for cancer prevention.” The similar study has already published in 2020 as in reference 17.

6. PLOS authors have the option to publish the peer review history of their article (what does this mean?). If published, this will include your full peer review and any attached files.

Reviewer #1: No

Reviewer #2: No

---

## [Author Response · Author response to Decision Letter 0]

12 Nov 2021

Dears Editors,

Thank you very much for giving us the opportunity to submit a revised version of our manuscript entitled: “Incidence of cervical, breast and colorectal cancer in people living with HIV between 2010 and 2015”.

Please find attached a complete author response to reviewers’ letter, a revised manuscript with track changes, together with an unmarked version of the revised manuscript without tracked changes. 

The entire manuscript has been thoroughly reviewed by a professional English mother-tongue copyeditor with over 15 years’ experience in the field. Following this, please change the title as follows: “Incidence of cervical, breast and colorectal cancers between 2010 and 2015 in people living with HIV in France”.

We hope that this revised version will meet the criteria for publication in the PLOS ONE journal. We remain at your disposal to make any changes, which could further improve our manuscript.

With best regards,

Maria Patrizia Carrieri, on behalf of all co-authors

---

## [Editor Report · Decision Letter 1]

24 Nov 2021

Incidence of cervical, breast and colorectal cancers between 2010 and 2015 in people living with HIV in France

PONE-D-21-14748R1

Dear Dr. Carrieri,

We’re pleased to inform you that your manuscript has been judged scientifically suitable for publication and will be formally accepted for publication once it meets all outstanding technical requirements.

Kind regards,

Cristian Apetrei, MD, PhD

Academic Editor

PLOS ONE
---

## [Editor Report · Acceptance letter]

17 Mar 2022

PONE-D-21-14748R1 

Incidence of cervical, breast and colorectal cancers between 2010 and 2015 in people living with HIV in France 

Dear Dr. Carrieri:

I'm pleased to inform you that your manuscript has been deemed suitable for publication in PLOS ONE. Congratulations! Your manuscript is now with our production department. 

Kind regards, 

on behalf of

Dr. Cristian Apetrei 

Academic Editor

PLOS ONE